# Association between Total Sugar Intake and Metabolic Syndrome in Middle-Aged Korean Men and Women

**DOI:** 10.3390/nu11092042

**Published:** 2019-09-01

**Authors:** Eun Ha Seo, Hyesook Kim, Oran Kwon

**Affiliations:** 1Department of Clinical Nutrition Science, The Graduate School of Clinical Health Sciences, Ewha Womans University, 52, Ewhayeodae-gil, Seodaemun-gu, Seoul 03760, Korea; 2Department of Nutritional Science and Food Management, Ewha Womans University, 52, Ewhayeodae-gil, Seodaemun-gu, Seoul 03760, Korea

**Keywords:** total sugar, chronic disease, Korean Genome and Epidemiology Study (KoGES)

## Abstract

There is increasing evidence emerging that suggests high sugar intake may adversely increase the incidence of chronic diseases. However, there are only a few related studies in Korea. Based on the current Dietary Reference Intakes for Koreans, this study examined whether total sugar intake above 20% of the total energy was a risk factor for metabolic syndrome in middle-aged Korean adults. This cross-sectional study involved 7005 adults (3751 men and 3254 women) aged 40–69 years, who participated in the Korean Genome and Epidemiology Study (KoGES), a large community-based cohort study. Daily total sugar intake was estimated using a validated food frequency questionnaire. About 9% and 16% of the men and women, respectively, derived >20% of energy intake from total sugar. The males in this category had a significantly higher odds of obesity defined as having a BMI ≥ 25 (OR = 1.491, 95% CI = 1.162–1.914), low HDL-cholesterol (OR = 1.313, 95% CI = 1.038–1.660), and metabolic syndrome (OR = 1.332, 95% CI = 1.038–1.709) than those who received a lower proportion of energy intake from total sugar. These results suggest that high (>20%) energy intake from total sugar may be associated with an increased risk of metabolic syndrome in middle-aged Korean men.

## 1. Introduction

Unhealthy eating patterns and a growing prevalence of chronic diseases, including diabetes and obesity, are a global concern. Healthy food choice is closely linked with the quality of life and healthy life expectancy, as well as the burden of healthcare expenditure. Consequently, among researchers, the public, and policymakers alike, there is a strong desire to identify the risk factors of diet-related chronic diseases [1]. In accord with this pursuit, there is a growing interest in the role of dietary patterns in the etiology of chronic diseases. While there have been many studies on fat intake associated with chronic diseases, sugar consumption has become a major concern in relation to the development of cardiovascular and metabolic diseases.

Biochemically, sugars consist of monosaccharides (glucose, fructose, galactose) and disaccharides (sucrose, maltose, lactose), excluding oligosaccharides [2]. Sugars occur naturally in some food, such as fruits and vegetables [3], in co-existence with other micronutrients and, therefore, have nutritional value [4]. By contrast, added sugars, which include refined sugars, honey, syrups, and other caloric sweeteners added to food or meals during processing or at home [2], provide no nutritional benefits [4]. Sugar-sweetened beverages (SSB) include sodas, sweetened juices, energy drinks, teas, and coffee that have been sweetened with added sugar [2,5].

Excessive intake of energy-dense sugary foods and beverage is linked to increased calorie intake and progressive body weight gain. Notably, high fructose consumption has been proposed as casually linked to obesity, diabetes, and hypertension, which besides, hypertriglyceridemia, low-HDL levels, and insulin resistance, are risk factors for metabolic syndrome [6]. Although the rationale that dietary sugar (particularly fructose) might cause or contribute to the disease process is theoretically based, it has raised debate around the amounts and thresholds of sugar consumption.

In 2015, the World Health Organization (WHO) recommended that countries committed to reducing the burden of non-communicable diseases should limit the intake of “free sugar” (added sugar, as well as sugars naturally present in fruit juice and fruit concentrates) in children and adults to less than 10% of the total energy intake (TEI) and, ideally, to below 5% for additional health benefits [7]. In the United States (US), the 2010 Dietary Guidelines Advisory Committee recommended that solid fat and added sugars should account for no more than 5–15% of the TEI [8], and in 2007, the United Kingdom (UK) proposed an average upper limit of 11% TEI from non-milk extrinsic sugars [9]. The 2015 Dietary Reference Intakes for Koreans (KDRI) direct a daily average sugar intake of 10–20% of the TEI (100 g based on 2000 kcal), and an added sugar (e.g., sugar, liquid fructose, starch syrup, molasses, honey, syrup, concentrated fruit juice) intake of within 10% of the TEI (50 g per 2000 kcal) [10]. These guidelines were justified by the fact that, when the percentage of TEI from total sugar is below 20%, as the total sugar intake increases, the intake of major nutrients also increases [11]. Meanwhile, for individuals with above 20% TEI from sugar, as the total sugar intake increases, the intake of some nutrients, such as protein, fat, sodium, and niacin, decreases [11] and the risk of metabolic syndrome increases [12].

According to the data estimating the consumption of sugars in various countries, daily mean total sugar intake for adults were 118.5 g in the US from 2009 to 2010, 95.1 g in the UK from 2008 to 2012, 104.8 g in Australia from 2011 to 2012, and 87.8 g in Sweden from 2010 to 2011 [13]. In Korea, daily mean total sugar intake from 2008 to 2011 was estimated as 61.4 g (percentage of energy from total sugar was 12.8%) according to the Korean National Health and Nutrition Examination Survey (KNHANES) [11]. When compared with western countries, total sugar intake among Koreans are relatively lower. However, the average daily total sugar intake of Koreans have gradually increased from 69.9 g in 2010 to 76.9 g in 2015, and when men and women were compared, the men’s intake was relatively higher (81.4 g in men and 72.4 g in women) [14].

Although the sugar intake of Koreans is not higher than that of western countries, over the past decades, the frequency of eating out and the intake of high sugar levels and processed food which are the characteristics of a Western diet, have been increasing [15,16] and are coincident with an increased prevalence of metabolic syndrome [17]. Around 27% of the entire Korean population are reported to have metabolic syndrome (30% in men and about 25% in women) [18]. Hence, careful attention should be paid to sugar intake in Korea. Moreover, the difference in the prevalence of metabolic syndrome between sexes [19] highlights the importance of examining the relationship between total sugar intake and the risk of metabolic syndrome in adult men and women, especially those at risk of chronic diseases; yet there is a lack of related research in Korea. Therefore, this study examined whether a total sugar intake above 20% of the TEI (based on the current KDRI recommended level for total sugar intake) is a risk factor for metabolic syndrome in middle-aged Korean adult men and women who participated in the 2001–2002 Korean Genome and Epidemiology Study (KoGES).

## 2. Materials and Methods

### 2.1. Study Population

This study performed a cross-sectional analysis of the 2001–2002 data provided by the KoGES, a prospective, community-based cohort study that started in 2001 and since then, conducts biennial follow-up studies. The KoGES analyzes the effect of lifestyle, intake, and environment on the incidence of chronic disease in participants aged 40–69 years living in community residences. Two South Korean communities were selected, one from Ansung, representing a rural community, and the other from Ansan, representing an urban community. Detailed information on the study procedure was described in a previous report [20].

Among 10,030 participants, those diagnosed with myocardial infarction, stroke, coronary artery disease, congestive heart failure, and history of cancer were excluded (*n* = 436). The following participants were also excluded due to an incomplete food frequency questionnaire (FFQ) (*n* = 691), unreliable energy intake of ≤500 kcal or >5000 kcal (*n* = 76), missing data regarding metabolic syndrome-related indicators (*n* = 243), and missing covariates, such as menopausal status (*n* = 1,579). Finally, a total of 7005 subjects (3751 men and 3254 women) were included for this cross-sectional analysis.

### 2.2. General Characteristics

Individual questionnaires, including lifestyle and disease history questions, were filled out by all participants. Residential location was divided as Ansan and Ansung. Three levels of education (elementary school, middle school/high school, and college/graduate school), alcohol consumption (non-drinker, ex-drinker, current drinker), and smoking behavior (non-smoker, ex-smoker, current smoker) were formed. Physical activity was defined as none or more than 30 min/day. Menopausal status was defined as “yes” or “no”, depending on whether menstruation had occurred for 12 months or more.

### 2.3. Measurement of Anthropometry and Metabolic Parameters

Anthropometry and metabolic parameters were measured by trained medical staff, with subjects wearing light clothes. Body mass index (BMI) was calculated as weight (kg) divided by the square of the height (m^2^). Blood pressure measurement was conducted at least twice under comfortable conditions. Blood samples obtained from the participants who fasted for 8 h were analyzed for biochemical markers, such as HDL-cholesterol, triglycerides, fasting blood glucose, insulin, C-reactive protein, and glycosylated hemoglobin. The homeostatic model assessment of insulin resistance ([fasting insulin × fasting glucose]/405) [21], LDL-cholesterol (total cholesterol—HDL-cholesterol—[triglycerides]/5]) [21], and atherogenic index ([total cholesterol—HDL-cholesterol]/HDL-cholesterol) [22] were calculated as indicated.

### 2.4. Definitions of Metabolic Syndrome

Metabolic syndrome was diagnosed using two approaches: The Adult Treatment Panel III of the National Cholesterol Education Program’s (NCEP ATP III) standard diagnostic criteria [23], and the Korean Society for the Study of Obesity guidelines on waist circumference (WC) for defining abdominal obesity [24]. Metabolic syndrome was defined as having three or more of the following:WC: Male > 90 cm, female > 85 cmSystolic blood pressure ≥ 130 mmHg, diastolic blood pressure ≥ 85 mmHgFasting blood glucose ≥ 100 mg/dLTriglycerides ≥ 150 mg/dLHDL-cholesterol: Male < 40 mg/dL, female < 50 mg/dL

### 2.5. Dietary Assessment

Dietary intake was assessed using a validated 103-item FFQ. All study procedures were conducted by trained interviewers. Participants were asked to report the usual frequency and portion size of each food consumed during the past year. The answer for frequency had nine options for each food: “never/almost,” “once/month,” “2–3 times/month,” “1–2 times/week,” “3–4 times/week,” “5–6 times/week,” “once/day,” “2 times/day,” and “≥3 times/day.” The answer for portion size had three options for each food: “small (1/2 serving/day),” “standard (1 serving/day),” and “large (≥2 servings/day).” The portion size of each food was determined based on the KoGES FFQ guideline. For analysis, food consumption was converted to weekly frequencies and then multiplied by the reported portion sizes.

Dietary nutrient intake data were compared using the estimated energy requirement and the estimated average requirement of the 2015 KDRI [10].

#### Estimation of Total Sugar Intake

Daily total sugar intake was estimated from the total sugar database constructed by the Korea Food and Drug Administration (KFDA) [25] for the foods included in the FFQ recipe. The percentage of energy from sugar intake was calculated as daily total sugar intake × 4 kcal/energy intake × 100.

### 2.6. Statistical Analysis

Data were expressed as mean ± standard deviation for continuous variables and number (*n*, %) for categorical variables. To identify the significance of the difference in means and distribution of general characteristics and dietary intakes between the two groups (≤20% TEI from total sugar vs. >20% TEI from total sugar), the Student’s *t*-test (for continuous variables: Age, BMI) and the chi-square test and Fisher’s exact test (categorical variables: residential location, educational level, alcohol consumption, smoking behavior, physical activity, menopausal status) were used.

Differences in metabolic parameters between the two groups were evaluated using the general linear model (GLM) test after adjusting for potential confounders. The potential confounders included the following variables: Age, energy intake, residential location, educational level, alcohol consumption, smoking behavior, physical activity, and menopausal status.

Multiple logistic regression analysis was performed to estimate the odds ratios (OR) and 95% confidence intervals (CI) for cardiometabolic biomarkers and metabolic syndrome, depending on TEI from total sugar (≤20% vs. >20%). Potential confounders were included in the GLM and multiple logistic regression analysis as covariates. All statistical analyses were performed using SAS software version 9.4. Statistical significance was considered when *p* < 0.05.

## 3. Results

### 3.1. General Characteristics of the Subjects According to the Percentage of Energy from Total Sugar Intake

The general characteristics of the two groups (≤20% TEI from total sugar vs. >20% TEI from total sugar) are displayed in Table 1. Mean total sugar intakes and mean percentage TEI from total sugar of men and women were 66.5 ± 40.6 and 68.0 ± 45.7 g/day and 12.8 ± 5.4% and 14.4 ± 6.7%/day, respectively (data not shown). About 9% of the men and 16% of the women derived >20% TEI from total sugar. The males in this category had a lower proportion of current drinkers, a higher proportion of current smokers, and were significantly older than the males who received ≤20% TEI from total sugar. Regarding residential location, the proportions of men and women who had >20% TEI from total sugar were significantly higher in Ansung than Ansan (*p* < 0.0001). In women, there was no difference in general characteristics between the two groups (≤20% TEI from total sugar vs. >20% TEI from total sugar), except for residential location.

### 3.2. Daily Nutrient Intake According to the Percentage of Energy from Total Sugar Intake

In men and women, daily intakes of all macronutrients, including energy, were significantly higher in the group with >20% TEI from total sugar than the group with ≤20% TEI from total sugar (Table 2). Vitamin and mineral intakes showed the same trend (data not shown). When comparing the two groups as contribution percentage in the energy intake (i.e., not the absolute intake of macronutrients), the percentage of energy intake from carbohydrate in men, and the percentage of energy intake from fat and carbohydrate in women were significantly higher in the group with >20% TEI from total sugar.

### 3.3. Metabolic Parameters According to the Percentage of Energy from Total Sugar Intake

Men who received >20% TEI from total sugar had higher WC (*p* = 0.0481) and lower HDL-cholesterol (*p* = 0.0067) than those who derived ≤20% TEI from total sugar (Table 3). This trend was also evident in women (*p* = 0.0455).

### 3.4. Association between the Percentage of Energy from Total Sugar Intake and Cardiometabolic Biomarkers Risk

Men with >20% TEI from total sugar had a higher odds of obesity (defined as BMI ≥ 25 kg/m^2^) (OR = 1.491, 95% CI = 1.162–1.914) after adjusting for covariates (Table 4) in comparison to the men with ≤20% TEI from total sugar Except for obesity, there was no difference in the odds of other cardiometabolic biomarker risk factors between the two groups of men. There was no difference in the odds of all cardiometabolic biomarker risk factors between the two groups of women.

### 3.5. Association between the Percentage of Energy from Total Sugar Intake and Metabolic Syndrome

Table 5 shows that men who derived >20% TEI from total sugar had a higher odds of low HDL-cholesterol (OR = 1.313, 95% CI = 1.038–1.660) and metabolic syndrome (OR = 1.332, 95% CI = 1.038–1.709) when compared with those who received ≤20% TEI from total sugar There was no difference in odds of metabolic syndrome between the two groups of women.

## 4. Discussion

This study found that deriving >20% TEI from total sugar was associated with higher odds of obesity, low HDL-cholesterol, and metabolic syndrome among Korean men aged 40–69 years, but not among women in the same age range.

The study of the relationship between sugar intake and chronic diseases has been mainly focused on SSB, with insufficient analysis of total sugar intake. In addition, investigations into the association between total sugar intake and chronic diseases, including metabolic syndrome, in adults who are not Korean adolescents are even rarer. Recently, the KFDA reported the results of a study examining the relationship between the total sugar intake (calculated using two-day 24-h recall data) and chronic diseases (using the KoGES data). It found that the risk of obesity, diabetes, and hypertension was not affected by whether total sugar intake was above or below 20% TEI. However, the risk of overweight was higher in men who derived >20% TEI from total sugar compared with those who received <20% TEI from total sugar [26].

The proportion of processed foods (56.8%) contributing to total sugar intake of Koreans was higher than that from food in which sugars occur naturally, and the top five processed food sources of total sugar intake were granulated sugar, carbonated beverages, coffee, bread, and fruit and vegetable drinks [11]. In other words, among processed foods, SSB were found to be the main source of sugar intake. It has been reported that SSB negatively affect health because they generally do not give a feeling of fullness and only increase energy density without providing other nutrients. Shin et al. [27] argued that high consumption of SSB was closely linked to a higher prevalence of obesity and metabolic syndrome in the Korean population. Analysis by Kwak et al. [28] demonstrated that among 5575 middle-aged participants in the KoGES, those in the highest SSB consumption group had a 21% greater risk of hypertension than those in the lowest SSB consumption group, and this association was most relevant for the participants with obesity. This relationship between SSB and chronic diseases has also been reported in Asian [29,30] and Western [31,32] populations.

There are several possible mechanisms explaining the link between SSB intake and chronic disease. First, the energy consumed in the form of SSB is rapidly absorbed, resulting in a positive energy balance, which, in turn, leads to increased weight since more energy is consumed in the next meal [33]. Second, apart from the weight gain effect, SSB intake increases the dietary glycemic load and fructose intake, which increases insulin resistance, beta-cell function, inflammation, hypertension, accumulation of visceral fat, and, eventually, increases the risk of metabolic syndrome, type 2 diabetes, and cardiovascular disease [34]. Moreover, artificial sweeteners have also been reported to enhance the risk of glucose intolerance. Continued consumption of artificial sweeteners can alter the composition and function of the gut microbiota, which may lead to the induction of metabolic abnormalities, such as glucose intolerance [35].

Interestingly, our study found a positive association between total sugar intake and metabolic syndrome in men, but not in women, even though the proportion of the highest consumption group in men was lower than that in women. It is unclear why only men showed a positive association between total sugar intake and metabolic disease. Different outcomes in the association of total sugar or SSB with metabolic disease between sexes can be found in the literature. In a cross-sectional study, SSB intake was significantly associated with obesity in men, but metabolic syndrome and obesity in women [27]. In other work, a positive association between total sugar intake and overweight was exclusively observed in men [26]. When examining the data of 5797 middle-aged Korean men and women, Kang and Kim [36] revealed that frequent soft drink consumption (>4 servings/week) was associated with an increased risk of developing metabolic syndrome and its components only in women. Elsewhere, women, but not men, who consumed ≥1 SSB/day had an increased incidence of type 2 diabetes [30]. These sex-dependent results might be explained by dietary intake and lifestyle differences between men and women that could alter hormone levels and metabolic profile, which could affect obesity or metabolic disease.

In the present study, in men, but not women, there was a significant association between high sugar intake and unhealthy lifestyle (such as smoking). High sugar consumption (particularly as SSB) is a known marker of an unhealthy lifestyle. Long-term high intake of soft drinks (≥3 times per week) has been significantly associated with a lower level of physical activity and a greater degree of smoking than long-term, low intake of soft drinks (<3 times per week) [37]. Another study found that high soft drink intake (≥ twice a day) was associated with more smoking than low soft drink intake (< once a day) [32].

Low estrogen concentration causes low HDL-cholesterol and leads to resistance to insulin-mediated glucose [38,39] because estrogen protects HDL-cholesterol by suppressing hepatic lipase and is tightly associated with lipoprotein profiles [40,41]. Since these processes contribute to obesity and type 2 diabetes, they increase the risk of exposure to metabolic syndrome and cardiovascular disease risk factors [33,42]. Therefore, women maintaining an HDL-cholesterol level at 10 mg/dL higher than men by estrogen have a relatively lower risk of metabolic syndrome and cardiovascular disease [43]. As a result, differences by sex can be interpreted according to evidence that estrogen’s effect on lipid metabolism and lifestyle may contribute to the association between sugar intake and metabolic syndrome.

In our study, participants who exceeded 20% TEI from total sugar had higher nutrient intake than those with ≤20% TEI from total sugar. These results may be caused by differences in the major sources of sugar, highlighting the importance of measuring sugar intake, considering the main dietary source of sugar. Physiological metabolic pathways are affected by the form of sugars. Unlike simple sugars that are absorbed in the upper half of the gastrointestinal tract, complex carbohydrates are absorbed after decomposition by microbes in the large intestine [44]. During this process, short-chain fatty acids are produced, stimulating enteroendocrine pathways, thereby increasing the levels of satiety hormones [44]. In comparing solid intake with liquid intake, the masticatory movement of eating solid food triggers satiety hormones, which can reduce total energy intake [45]. Furthermore, pancreatic exocrine and endocrine mechanisms that act to release insulin function better, and this can enhance the regulation of glucose tolerance [45]. For these reasons, even with isoenergetic sugar intake, the outcomes on health appear different depending on the original form of the food.

This study had several limitations. First, because our data are based on a cross-sectional study, it is difficult to explain the causal relationship between sugar intake and metabolic syndrome. Therefore, a further prospective cohort study is needed. Second, it is difficult to generalize the results to the entire Korean population because we only selected participants in Ansan and Ansung. Therefore, the results should be interpreted carefully. Although the study had some limitations, we analyzed the data after controlling for potential confounders to reduce the impact of these limitations. To the best of our knowledge, this is the first study to examine the associations of total sugar intake with obesity and metabolic syndrome (including its discrete components) at the same time, among middle-aged Korean adults. In particular, it is important to examine total sugar intake as a contribution to energy rather than total sugar intake in relation to obesity or metabolic syndrome.

Unlike the ongoing research activities in the West, research on sugar intake and chronic diseases in Korea is at the beginning stage. Korea and the West have different diets, and so it is reasonable to propose that there will be differences in their vulnerability to chronic diseases in relation to their diet. This study examined the relevance of total sugar intake as a risk factor of metabolic syndrome in middle-aged Korean adults. It sets a precedent for further research in this population that will confirm or not the suggestive evidence provided in this investigation that found total sugar intake is significantly associated with metabolic syndrome in middle-aged Korean men (40–69 years) but not women. Furthermore, the results of this study can be used as a scientific basis for setting the target level in order to gradually reduce not only the sugar intake of Koreans but populations worldwide. The data may also be used to suggest the appropriate policy for reducing the sugar intake of middle-aged adults.

## 5. Conclusions

In conclusion, this study found that total sugar intake was significantly associated with metabolic syndrome in middle-aged Korean men (40–69 years). Total sugar intake exceeding >20% of the TEI was associated with higher risks of low HDL-cholesterol, obesity (defined as BMI ≥25 kg/m^2^), and metabolic syndrome in men. While total sugar intake is important in terms of its contribution to the ratio of energy intake, other added-sugar/free-sugar intake levels and food sources are important and their causal relationship with metabolic syndrome should be identified using this comprehensive sugar intake evaluation.

## Figures and Tables

**Table 1 nutrients-11-02042-t001:** General characteristics of participants according to the percentage of energy from total sugar intake ^1^.

Characteristic	Men (*n* = 3751)			Women (*n* = 3254)		
	%Energy from Total Sugar ≤20 (*n* = 3404)	%Energy from Total Sugar >20 (*n* = 347)	*p*-Value ^2^	%Energy from Total Sugar ≤20 (*n* = 2734)	%Energy from Total Sugar >20 (*n* = 520)	*p*-Value ^2^
Age (years)	51.0 ± 8.6	52.4 ± 8.9	0.0047	53.3 ± 9.1	53.5 ± 9.0	0.6515
BMI (kg/m^2^)	24.2 ± 2.9	24.4 ± 3.1	0.4813	24.9 ± 3.3	25.1 ± 3.2	0.2371
Residential location			<0.0001			<0.0001
Ansung	1251 (36.8)	214 (61.7)		1351 (49.4)	338 (65.0)	
Ansan	2153 (63.3)	133 (38.3)		1383 (50.6)	182 (35.0)	
Educational level			0.1448			0.5457
Below elementary school	635 (18.7)	76 (21.9)		1269 (46.4)	244 (46.9)	
Middle school/high school	2025 (59.5)	188 (54.2)		1294 (47.3)	250 (48.1)	
College/graduate school	744 (21.9)	83 (23.9)		171 (6.3)	26 (5.0)	
Alcohol consumption			0.0209			0.1718
Non-drinker	600 (17.6)	81 (23.3)		1989 (72.8)	368 (70.8)	
Ex-drinker	316 (9.3)	35 (10.1)		75 (2.7)	22 (4.2)	
Current drinker	2488 (73.1)	231 (66.6)		670 (24.5)	130 (25.0)	
Smoking behavior			0.0126			0.5062
Non-smoker	661 (19.4)	72 (20.8)		2604 (95.3)	501 (96.4)	
Ex-smoker	1074 (31.6)	83 (23.9)		35 (1.3)	6 (1.2)	
Current smoker	1669 (49.0)	192 (55.3)		95 (3.5)	13 (2.5)	
Physical activity ^3^			0.2561			0.1902
No	112 (3.3)	16 (4.6)		13 (0.5)	5 (1.0)	
Yes	3292 (96.7)	331 (95.4)		2721 (99.5)	515 (99.0)	
Menopausal status						0.6007
No				1104 (40.4)	203 (39.0)	
Yes				1630 (59.6)	317 (61.0)	

^1^ Data are expressed as mean ± standard deviation or *n* (%). ^2^ Student’s *t*-test for continuous variables, and chi-square or Fisher’s exact test for categorical variables were used to analyze *p*-value. ^3^ Physical activity was defined as “yes” or “no”, depending on whether or not exercise was performed for ≥30 min/day. BM1, body mass index.

**Table 2 nutrients-11-02042-t002:** Daily intakes of energy and nutrient in the subjects according to the percentage of energy from total sugar intake ^1^.

Intake Parameter	Men (*n* = 3751)			Women (*n* = 3254)		
	%Energy from Total Sugar ≤20 (*n* = 3404)	%Energy from Total Sugar >20 (*n* = 347)	*p*-Value ^2^	%Energy from Total Sugar ≤20 (*n* = 2734)	%Energy from Total Sugar >20 (*n* = 520)	*p*-Value ^2^
Energy (kcal/day)	1982.5 ± 550.2	2303.9 ± 752.0	<0.0001	1819.7 ± 603.6	2071.9 ± 712.6	<0.0001
Protein (g/day)	68.6 ± 25.0	78.1 ± 29.1	<0.0001	60.6 ± 24.6	69.0 ± 27.9	<0.0001
Fat (g/day)	35.3 ± 18.1	40.9 ± 20.3	<0.0001	27.3 ± 16.9	32.3 ± 17.3	<0.0001
Carbohydrate (g/day)	341.3 ± 88.9	408.6 ± 136.2	<0.0001	326.9 ± 105.9	380.0 ± 133.0	<0.0001
%Energy from protein	13.7 ± 2.3	13.6 ± 2.3	0.4019	13.2 ± 2.3	13.3 ± 2.6	0.2019
%Energy from fat	15.4 ± 5.0	15.9 ± 5.2	0.1322	13.0 ± 5.2	14.0 ± 5.4	0.0002
%Energy from carbohydrate	69.5 ± 6.5	71.0 ± 6.9	<0.0001	72.4 ± 6.6	73.4 ± 7.5	0.0062

^1^ Data are expressed as mean ± standard deviation. ^2^ Student’s *t*-test was used to analyze *p*-value.

**Table 3 nutrients-11-02042-t003:** Metabolic parameters of participants according to the percentage of energy from total sugar intake ^1^.

Metabolic Parameter	Men (*n* = 3751)			Women (*n* = 3254)		
	%Energy from Total Sugar ≤20 (*n* = 3404)	%Energy from Total Sugar >20 (*n* = 347)	*p*-Value ^2^	%Energy from Total Sugar ≤20 (*n* = 2734)	%Energy from Total Sugar >20 (*n* = 520)	*p*-Value ^2^
WC (cm)	83.4 ± 7.5	84.7 ± 7.9	0.0481	81.6 ± 9.5	83.4 ± 9.5	0.0455
SBP (mmHg)	124.6 ± 17.4	125.3 ± 17.4	0.6158	124.0 ± 19.9	126.0 ± 19.4	0.3130
DBP (mmHg)	83.2 ± 11.3	83.1 ± 11.2	0.5797	80.3 ± 12.0	81.1 ± 11.5	0.9307
FBS (mg/dL)	90.7 ± 23.9	87.6 ± 21.2	0.1702	85.3 ± 18.8	84.6 ± 19.9	0.5972
Triglycerides (mg/dL)	176.6 ± 118.5	178.6 ± 109.0	0.8034	147.7 ± 86.9	148.0 ± 81.0	0.9788
HDL-cholesterol (mg/dL)	43.9 ± 10.0	42.2 ± 9.2	0.0067	45.7 ± 10.0	45.1 ± 9.4	0.2994
No. of MS components	1.7 ± 1.2	1.8 ± 1.2	0.0630	1.9 ± 1.3	2.0 ± 1.3	0.8220
Insulin (µIU/mL)	7.0 ± 4.1	7.2 ± 4.5	0.9684	8.0 ± 5.2	8.4 ± 4.6	0.8490
HbA1c (%)	5.8 ± 0.9	5.8 ± 0.9	0.5023	5.7 ± 0.8	5.8 ± 0.8	0.2990
HOMA-IR	1.6 ± 1.1	1.6 ± 1.1	0.6196	1.7 ± 1.2	1.8 ± 1.1	0.9646
CRP (mg/dL)	0.2 ± 0.5	0.2 ± 0.4	0.7928	0.2 ± 0.4	0.2 ± 0.5	0.2783
LDL-cholesterol (mg/dL)	113.9 ± 35.7	109.7 ± 31.4	0.9834	117.0 ± 32.6	114.2 ± 29.9	0.2525
AI	3.6 ± 1.2	3.6 ± 1.2	0.1085	3.4 ± 1.1	3.3 ± 1.1	0.8657

^1^ Data are expressed as mean ± standard deviation. ^2^ General linear model (GLM) was used to analyze *p*-value after adjusting for age (years), energy intake (kcal), residential location (Ansung and Ansan), educational level (elementary school, middle school/high school, college/graduate school), alcohol consumption (non-drinker, ex-drinker, current drinker), smoking behavior (non-smoker, ex-smoker, current smoker), physical activity (in days; none or ≥30 min/day), and menopausal status (“yes” or “no”; only women). WC, waist circumference; SBP, systolic blood pressure; DBP, diastolic blood pressure; FBS, fasting blood glucose; MS, metabolic syndrome; HbA1c, glycosylated hemoglobin; HOMA-IR, homeostasis model assessment estimate of insulin resistance; CRP, c-reactive protein; AI, atherogenic index.

**Table 4 nutrients-11-02042-t004:** Multivariate odds ratios (OR) and 95% confidence intervals (CI) of cardiometabolic biomarkers risk according to the percentage of energy from total sugar intake ^1^.

Cardiometabolic Biomarker	Men (*n* = 3751)		Women (*n* = 3254)	
	%Energy from Total Sugar ≤20 (*n* = 3404)	%Energy from Total Sugar >20 (*n* = 347)	%Energy from Total Sugar ≤20 (*n* = 2734)	%Energy from Total Sugar >20 (*n* = 520)
Obesity ^2^				
No. of case (%)	1149 (33.8)	128 (36.9)	971 (35.5)	188 (36.2)
Multivariate OR (95% CI)	1.000 (ref)	1.491 (1.162–1.914)	1.000 (ref)	1.143 (0.932–1.401)
High insulin ^3^				
No. of case (%)	822 (24.2)	82 (23.6)	903 (33.0)	204 (39.2)
Multivariate OR (95% CI)	1.000 (ref)	0.945 (0.723–1.237)	1.000 (ref)	1.162 (0.953–1.417)
High HbA1c ^4^				
No. of case (%)	298 (8.8)	27 (7.8)	228 (8.3)	40 (7.7)
Multivariate OR (95% CI)	1.000 (ref)	0.906 (0.592–1.385)	1.000 (ref)	0.898 (0.626–1.287)
High HOMA-IR^5^				
No. of case (%)	842 (24.7)	80 (23.1)	740 (27.1)	159 (30.6)
Multivariate OR (95% CI)	1.000 (ref)	0.919 (0.701–1.204)	1.000 (ref)	1.075 (0.871–1.327)
High CRP ^6^				
No. of case (%)	92 (2.7)	11 (3.2)	54 (2.0)	12 (2.3)
Multivariate OR (95% CI)	1.000 (ref)	1.263 (0.654–2.436)	1.000 (ref)	1.127 (0.590–2.152)
High LDL-cholesterol ^7^				
No. of case (%)	1100 (32.3)	91 (26.2)	885 (32.4)	147 (28.3)
Multivariate OR (95% CI)	1.000 (ref)	0.908 (0.697–1.182)	1.000 (ref)	0.850 (0.685–1.054)
High AI ^8^				
No. of case (%)	371 (10.9)	36 (10.4)	211 (7.7)	35 (6.7)
Multivariate OR (95% CI)	1.000 (ref)	0.999 (0.684–1.457)	1.000 (ref)	0.879 (0.601–1.287)

^1^ Multiple logistic regression was used to estimate OR (95% CI) after adjusting for age (years), energy intake (kcal), residential location (Ansung and Ansan), educational level (elementary school, middle school/high school, college/graduate school), alcohol consumption (non-drinker, ex-drinker, current drinker), smoking behavior (non-smoker, ex-smoker, current smoker), physical activity (in days; none or ≥30 min/day), and menopausal status (“yes” or “no”; only women). ^2^ Obesity (BMI ≥ 25 kg/m^2^). ^3^ High insulin (≥9 µIU/mL). ^4^ High HbA1c (≥6.5%). ^5^ high HOMA-IR ([fasting glucose × fasting insulin]/405; >2). ^6^ High CRP (>1 mg/dL). ^7^ High LDL-cholesterol (≥130 mg/dL). ^8^ High AI ([total cholesterol—HDL-cholesterol]/HDL-cholesterol; >5). HbA1c, glycosylated hemoglobin; HOMA-IR, homeostasis model assessment estimate of insulin resistance; CRP, c-reactive protein; AI, atherogenic index.

**Table 5 nutrients-11-02042-t005:** Multivariate odds ratios (OR) and 95% confidence intervals (CI) of metabolic syndrome according to the percentage of energy from total sugar intake ^1^.

Metabolic Syndrome Parameter	Men (*n* = 3751)		Women (*n* = 3254)	
	%Energy from Total Sugar ≤20 (*n* = 3404)	%Energy from Total Sugar >20 (*n* = 347)	%Energy from Total Sugar ≤20 (*n* = 2734)	%Energy from Total Sugar >20 (*n* = 520)
Abdominal obesity ^2^				
No. of case (%)	606 (17.8)	81 (23.3)	948 (34.7)	213 (41.0)
Multivariate OR (95% CI)	1.000 (ref)	1.218 (0.926–1.604)	1.000 (ref)	1.129 (0.917–1.389)
High blood pressure ^3^				
No. of case (%)	1629 (47.9)	185 (53.3)	1165 (42.6)	237 (45.6)
Multivariate OR (95% CI)	1.000 (ref)	1.194 (0.945–1.509)	1.000 (ref)	1.011 (0.822–1.245)
High glucose ^4^				
No. of case (%)	535 (15.7)	35 (10.1)	235 (8.6)	33 (6.4)
Multivariate OR (95% CI)	1.000 (ref)	0.716 (0.494–1.038)	1.000 (ref)	0.755 (0.513–1.109)
High triglycerides ^5^				
No. of case (%)	1641 (48.2)	174 (50.1)	959 (35.1)	187 (36.0)
Multivariate OR (95% CI)	1.000 (ref)	1.121 (0.892–1.410)	1.000 (ref)	1.041 (0.848–1.277)
Low HDL-cholesterol ^6^				
No. of case (%)	1218 (35.8)	150 (43.2)	1930 (70.6)	367 (70.6)
Multivariate OR (95% CI)	1.000 (ref)	1.313 (1.038–1.660)	1.000 (ref)	0.979 (0.792–1.210)
Metabolic syndrome ^7^				
No. of case (%)	823 (24.2)	107 (30.8)	900 (32.9)	174 (33.5)
Multivariate OR (95% CI)	1.000 (ref)	1.332 (1.038–1.709)	1.000 (ref)	0.922 (0.743–1.143)

^1^ Analyzed by logistic regression for OR (95% CI) after adjusting for age (in years; 40–69), energy intake, residential location (Ansung and Ansan), educational level (elementary school, middle school/high school, college/graduate school), alcohol consumption (non-drinker, ex-drinker, current drinker), smoking behavior (non-smoker, ex-smoker, current smoker), physical activity (in days; none or ≥30 min/day), and menopausal status (“yes” or “no”; only women). ^2^ Abdominal obesity (WC: male > 90 cm, female > 85 cm). ^3^ High blood pressure (SBP ≥ 130 mmHg or DBP ≥ 85 mmHg). ^4^ High glucose (FBS ≥ 100 mg/dL). ^5^ High triglycerides (triglycerides ≥ 150 mg/dL). ^6^ Low HDL-cholesterol (HDL-cholesterol: male < 40 mg/dL, female < 50 mg/dL). ^7^ Metabolic syndrome (WC, triglycerides, blood pressure, FBS, HDL-cholesterol; ≥3 of 5). WC, waist circumference; SBP, systolic blood pressure; DBP, diastolic blood pressure; FBS, fasting blood glucose.

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
