# Peer review of "Association between Total Sugar Intake and Metabolic Syndrome in Middle-Aged Korean Men and Women"

_nutrients, 2019, doi:10.3390/nu11092042_

Round 1

Reviewer 1 Report

The present study aims to determine if the total sugar intake represents a risk factor for metabolic syndrome in middle-aged Korean adults. Therefore, authors performed a cross-sectional study involved over 7k adults aged between forties and seventies that have participated in the Korean Genome and Epidemiology Study and the amount of sugar consumption was estimated using a validated food intake survey.

This work is a relevant and exciting theme because is not only significant to Korean people but also represent an important global relevance.

The present study is very well designed from the methodological point of view, very well written and very well presented. It is easy to read and understand.

Even so, here I send some suggestions, that from my point-of-view could be addressed by the authors:

Point 1

Please avoid followed brackets. Eg.:

Line 83 “(≤500 kcal or >5000 kcal) (n = 76)”

Suggestion to improvement: Consider rephrase it.

Point 2

Please give some information about residential location such as Ansung or Ansan. For non-Koreans this is not understandable the importance to indicate these two locations. Are they different? Socio-economically? Socio-demographically? Geographically? This is what are the variables that are important / relevant enough to be mentioned?

Suggestion to improvement: Include some description in material and methods or consider removing it.

Point 3

Regarding to table 1, titles of the columns are “% Energy from total sugar ≤20 “, “% Energy from total sugar > 20 “, and so on. Since you mention that the values are in percentage, and the valued on brackets are the values of participants (n) the values of percentage should not be “hidden” in the brackets but should be the significant ones. For example:

… /…

Below elementary school          635 (18.7)

Suggestion to improvement:

… /…

Below elementary school          18.7 (635)

Please consider this comment to be applied to table 4 and 5.

Author Response

The present study aims to determine if the total sugar intake represents a risk factor for metabolic syndrome in middle-aged Korean adults. Therefore, authors performed a cross-sectional study involved over 7k adults aged between forties and seventies that have participated in the Korean Genome and Epidemiology Study and the amount of sugar consumption was estimated using a validated food intake survey.

This work is a relevant and exciting theme because is not only significant to Korean people but also represent an important global relevance.

The present study is very well designed from the methodological point of view, very well written and very well presented. It is easy to read and understand.

Even so, here I send some suggestions, that from my point-of-view could be addressed by the authors:

-> Authors: We sincerely appreciate your insightful and constructive comments and suggestions. Please see our detailed responses below.

Point 1

Please avoid followed brackets. Eg.:

Line 83 “(≤500 kcal or >5000 kcal) (n = 76)”

Suggestion to improvement: Consider rephrase it.

-> Authors: Following your suggestion, we removed the brackets and corrected the sentence. (Line 89).

Point 2

Please give some information about residential location such as Ansung or Ansan. For non-Koreans this is not understandable the importance to indicate these two locations. Are they different? Socio-economically? Socio-demographically? Geographically? This is what are the variables that are important / relevant enough to be mentioned?

Suggestion to improvement: Include some description in material and methods or consider removing it.

-> Authors: Thank you for these valuable comments. Following your suggestion, we explain our choice for the residential locations Ansung and Ansan (Lines 92–95).

Point 3

Regarding to table 1, titles of the columns are “% Energy from total sugar ≤20 “, “% Energy from total sugar > 20 “, and so on. Since you mention that the values are in percentage, and the valued on brackets are the values of participants (n) the values of percentage should not be “hidden” in the brackets but should be the significant ones. For example:

/…Below elementary school         635 (18.7)

Suggestion to improvement:

/…Below elementary school         18.7 (635)

Please consider this comment to be applied to table 4 and 5.

-> Authors: Table 1 compares the general details of the groups who consume less than 20% of total energy intake from total sugar and those who consume more than 20%. However, it does not mean that the values are expressed in % because they compare the characteristics (e.g., age, residential location, educational level) between the two groups. The values provided are therefore expressed as the mean ± standard deviation or n (%). The data in Tables 4 and 5 are expressed as the odds ratios and 95% confidence intervals. As a result, we did not modify the original presentation, which is usually the way the table compares categorical variables in two groups. We hope this explanation clarifies the reviewers’ concern.

Reviewer 2 Report

This study is interesting as it provides insights regards the association between sugar intake and metabolic syndrome in the Korean (Ansan and Ansung) middle-aged population. This adds to the literature by providing data and first insights in the potential need to limit the intake of total sugar to reduce the prevalence of metabolic syndrome in Korea. Yet, I feel that the paper has some limitations, and some parts are not well explained in my opinion.

It is not clear to me why the authors conducted a cross-sectional analysis, while a longitudinal analysis would be interesting to see whether this sugar intake actually predicts metabolic syndrome. In the introduction:

the first sentence doesn’t provide the context. It would benefit the reader to explain where the dietary patterns have changed into the western diet. The relevance of the first paragraph isn’t entirely clear I feel there should be more elaboration regards sugar intake being a risk factor for developing metabolic syndrome. Also why it is especially important to look into this association in Korea… Not clear why this study focuses only on this specific age group. Also referring to middle-aged Korean men and women is a bit misleading as the authors only focused on Ansan and Ansung. No justification is given for that as well.

In the methods:

In the data analyses section, it is not clear what is categorical and what is continuous; I do not wish to look them up one by one; should be stated there again. It mentions adjusting for cofounders, such as…. ; I feel that the authors need to state all the cofounders, not ‘such as’ Was a power analysis conducted to justify the sample size?

In the discussion, again, the whole relevance of this study is not entirely clear… it isn’t explicitly elaborated why it is important to look into this association.

Author Response

This study is interesting as it provides insights regards the association between sugar intake and metabolic syndrome in the Korean (Ansan and Ansung) middle-aged population. This adds to the literature by providing data and first insights in the potential need to limit the intake of total sugar to reduce the prevalence of metabolic syndrome in Korea. Yet, I feel that the paper has some limitations, and some parts are not well explained in my opinion.

-> Authors: We sincerely appreciate your insightful and constructive comments and suggestions. Please see our detailed responses below.

It is not clear to me why the authors conducted a cross-sectional analysis, while a longitudinal analysis would be interesting to see whether this sugar intake actually predicts metabolic syndrome. In the introduction:

the first sentence doesn’t provide the context. It would benefit the reader to explain where the dietary patterns have changed into the western diet. The relevance of the first paragraph isn’t entirely clear I feel there should be more elaboration regards sugar intake being a risk factor for developing metabolic syndrome.

-> Authors: We are interested in knowing whether sugar intake predicts metabolic syndrome. While a longitudinal analysis would be most appropriate to answer this research question, the follow-up rate of this cohort study until 2014 was around 59%, which was low, and there were a lot of data to check for accurate calculation of the person year associated with metabolic syndrome. Therefore, our research team decided to examine these relationships initially as a cross-sectional study, and then conduct follow-up analysis through subsequent data cleaning. The results of this cross-sectional study suggested a relationship between total sugar intake and metabolic syndrome, and we will examine this association in more detail through follow-up studies.

We have rewritten the first paragraph to provide a more general statement regarding dietary patterns and the incidence of chronic diseases, while highlighting the growing concern that sugar intake might be a risk factor for the progress of chronic diseases (Lines 30–36). Further detail was added to paragraph three, focusing on sugar intake being a potential risk factor for developing metabolic syndrome (Lines 45–50). Finally, information was added to paragraph six, which shows that the increase in metabolic syndrome in Korean adult men and women over the past few decades is coincident with the trend of increase in sugar consumption, processed foods and eating out among the Korean population (Lines 75–83).

Also why it is especially important to look into this association in Korea… Not clear why this study focuses only on this specific age group. Also referring to middle-aged Korean men and women is a bit misleading as the authors only focused on Ansan and Ansung. No justification is given for that as well.

-> Authors: The reasons why it is especially important to look into this association in Korea and why this study focuses only on this specific age group are described in the sixth paragraph of the Introduction section (Lines 75–83). As mentioned in the Materials and Methods section, the KoGES analyzes the effect of lifestyle, intake, and environment on the incidence of chronic disease in participants aged 40–69 years living in community residences. Two South Korean communities were selected, one from Ansung, representing a rural community, and the other from Ansan, representing an urban community (Lines 92–95). On the points of concern, we mentioned the following in the limitations of the Discussion section (Lines 317–319). It is difficult to generalize the results to the entire Korean population because we only selected participants in Ansan and Ansung. Therefore, the results should be interpreted carefully.

In the methods:

In the data analyses section, it is not clear what is categorical and what is continuous; I do not wish to look them up one by one; should be stated there again. It mentions adjusting for cofounders, such as…. ; I feel that the authors need to state all the cofounders, not ‘such as’ Was a power analysis conducted to justify the sample size?

-> Authors: Following your suggestion, we described in detail what is categorical and what is continuous (Lines 152–154). We also stated all the cofounders (Lines 156–158). We did not perform a power analysis to estimate the number of subjects because this study was conducted on all subjects, except those who did not have relevant data or met the exclusion criteria.

In the discussion, again, the whole relevance of this study is not entirely clear… it isn’t explicitly elaborated why it is important to look into this association.

-> Authors: We further describe the necessity and importance of this study in the last paragraph of the discussion section (Lines 326–336).